# OpenReview forum: "Discrete Adjoint Matching"
_ICLR.cc/2026/Conference — ICLR 2026 Poster_

### Official Review · Reviewer_uj3e · 2025-10-22

**Soundness:** 3
**Presentation:** 2
**Contribution:** 3
**Rating:** 6
**Confidence:** 4

**Summary:**

The paper under consideration proposes an algorithm to fine-tune a discrete generative model. The core idea is to develop a discrete analog of the Adjoint Matching method recently proposed by Domingo Enrich et al. for continuous models. To achieve this generalization, the authors obtain at Theorem 2.2 an equation for the adjoint discrete variable that mirrors the stochastic Maximum principle for the auxiliary stochastic control problem used in Domingo Enrich et al. The resulting algorithm (Algorithm 1) is then derived following closely the strategy used in continuous spaces, with some differences. For example, samples from the optimal law $p^{\star}$, which are required to leverage the adjoint equation, are approximated using importance weights (Prop 2.4). The paper is completed by further theoretical results that elucidate similarities with the continuous setting (Sec. 3) and a section on numerical experiments where the method is tested both on synthetic examples and more complex mathematical reasoning tasks.

**Strengths:**

The paper extends an algorithmical framework that has drawn quite some attention for continuous models to the discrete setting in a principled way. The numerical results appear quite convincing.

**Weaknesses:**

- There is no theoretical guarantees of convergence for the proposed method (as there is none for its continuous counterpart). This is a sever limitation, in my opinion.

* On the methodological side, AM in continuous state space requires the pre-trained model to be such that the initial and final states are independent. I find this a very strong requirement hard to satisfy in practice, unless the pre-trained model had been designed to have such property. If I am not mistaken the same limitation is present here. I encourage the authors to clarify this point.

**Questions:**

- It seems to me that in equation (6) $f(y)$ should be replaced by $f(y)-f(X_\tau)$



- The paper puts quite some emphasis on the fact that the derivation of Theorem 2.2 avoids "convoluted control-theoretic derivation adopted in original AM and providing a more general framework for adjoint-based estimators." But at the end of the day, it seems to me that equation (8) is a form of the Pontryagin maximum principle for jump process, and I don't feel that it would be so difficult to obtain following for example  the framework developed in Appendix F of [1] for the control of CTMCs. From this perspective, many aspects of (8), such as the multiplicative nature of the control, appear as rather natural than surprising. I believe that adding the statistical estimator perspective offered in this work is valuable, and I am happy to see it. However, offering also a control theoretical approach would make it even clearer why the method proposed here is the discrete counter part of AM which ultimately relies on equation (3), which is nothing but the Pontryagin maximum principle. Relegating the control-theoretic derivation of Theorem 2.2 to the appendix does not appear to be a good choice to me.

*[1]  Le-Tuyet-Nhi, P. H. A. M., Shariatian, D., Ocello, A., Conforti, G., & Durmus, A. O. Discrete Markov Probabilistic Models: An Improved Discrete Score-Based Framework with sharp convergence bounds under minimal assumptions. In Forty-second International Conference on Machine Learning.*

---

> ### Author Response · Authors · 2025-11-26
> **Response to uj3e (1/2)**
>
> We thank the reviewer for raising all valuable comments. We are excited that the reviewer recognized the principled foundation of our algorithmic framework, appreciated our contribution, and acknowledged effectieness of our empirical results. We believe DAM takes a significant step toward principled methods for fine-tuning discrete generative models. We try our best to resolve all raised concerns in the responses below.
>
> &nbsp;
>
> ---
>
> > **1: Discussions on convergence in continuous AM and DAM (W1)**
>
> - We thank the reviewer for raising this comment. As the reviewer conjectured, the original continuous AM (Eq 3) does not admit theoretical guarantees on convergence. This is because continuous AM relies primarily on a fixed-point relation in Eq 3; specifically, the continuous adjoint $\tilde{a}_t(X)$ in Eq 3 is an unbiased estimator of the optimal control only when $X \sim p^\star$ is sampled from the optimal controlled distribution. This mutual dependency prevents straightforward convergence analysis of the original continuous AM.
> - While our initially derived discrete adjoint (Eq 8) exhibits a similar fixed-point form—and might therefore suffer from similar issues at first glance—we stress that **DAM does not implement this fixed-point relation**, as was done in the original continuous AM. Instead, DAM employs the importance-weighted discrete adjoint in Eq 12, which **uses SNIS to directly estimate the expectations**. Since SNIS is a consistent estimator [A], the importance-weighted discrete adjoint, $\hat{a}_t(X)$ with $X \sim p^u$ sampled from the model, converges to the true discrete adjoint as the number of model samples $K$ approaches infinity. We hope the reviewer recognizes the distinction.
> - Finally, we note that there are concurrent works which establish preliminary convergence guarantees for discrete generative models with the aid of trust region regularization (*e.g.*, Prop 3.1 in [B]). Incorporating insights from these studies would be a promising future direction.
>
> &nbsp;
>
> ---
>
> > **2: Memoryless requirement for DAM (W2)**
>
> - We first thank the reviewer for rasing the comments. As conjectured by the reviewer, DAM indeed requires the *memoryless* condition (*i.e.*, independent initial and final states). This is stated explicitly in the Introduction section (L53):
>   > ${}^1$ Formally, we require the base distribution to satisfy the memoryless condition, $p^{\text{base}}(X_1|X_0) = p^{\text{base}}(X_1)$,
> which readily holds for all base models considered.
>
>   Nevertheless, we recognize that this condition may be unclearly presented in the footnote, which readers may overlook. We have clarified this point in the revision (L37-38).
> - While we agree with the reviewer that the memoryless requirement for pretrained models may seem restrictive for general applications, we would like to note that, in the language generation setting—which is the primary focus of this work—*all* pretrained models predominantly satisfy the memoryless condition. This is partly due to the lack of informative prior for sampling $X_0$, and is primarily supported by empirical evidence: setting $X_0$ to the masked absorbing state—which readily fulfill the memory condition $p^{\text{base}}(X_1|X_0) = p^{\text{base}}(X_1)$—consistently outperforms other alternatives [C].
> - That said, we agree that for more general control settings, relaxing the memoryless condition is an important next step. Although this direction is beyond the scope of the current paper, we note that such relaxation has recently been explored for the continuous case in [D], and we believe it could be extended to the discrete case by introducing a second network to learn the correction term for the rewards. We thank the reviewer for bringing up this exciting future direction.
>
> &nbsp;
>
> ---
>
> > **3: Clarification in Dynkin's formula (Q1)**
>
> - We believe the current expression Dynkin's formula in Eq 7 is valid. To illustrate, note that the **summation is over all y** in Eq 7, *i.e.*,
> $$
> \sum_y u_\tau(y, X_\tau) f_\tau(y)
> $$
> In particular, this includes the term $u_\tau(X_\tau, X_\tau) = - \sum_{y \not= X_\tau} u_\tau(y, X_\tau)$. Rewriting this as a summation over only $y \not= X_\tau$ by splitting out the $y=X_\tau$ term would result in the $f(y) - f(X_\tau)$ the reviewer has mentioned, i.e.,
> $$
> \sum_{y \not= X_\tau} u_\tau(y, X_\tau) f_\tau(y) + u_\tau(X_\tau, X_\tau) f_\tau(X_\tau) = \sum_{y \not= X_\tau} u_\tau(y, X_\tau) \Big( \color{red}{f(y) - f(X_\tau)} \Big)
> $$
> That said, both expressions are equivalent and hence valid.

---

> > ### Comment · Reviewer_uj3e · 2025-11-26
> >
> > Thank you for clarifying this point. I agree with you that the original formula is correct.

---

> > ### Comment · Reviewer_uj3e · 2025-11-26
> >
> > I thank the authors for their detailed answers. While the lack of theoretical guarantees remains an issue, the argument they bring to justify the interest of a retrained model with independent coupling the masked state is relatively convincing, though I am not an expert of the implementation aspect of discrete diffusion models.
> >
> > In light of this improvement, I will slightly increase my score.

---

> ### Author Response · Authors · 2025-11-26
> **Response to uj3e (2/2)**
>
> > **4: Discussion on control-theoretic perspective (Q2)**
>
> - We thank the reviewer for highlighting the connection between (D)AM and the Pontryagin maximum principle. It is exciting to see that a similar mathematical framework—albeit applied to different domains (control vs. language modeling)—has been investigated independently. To the best of our understanding, this connection was likely not recognized in the original AM, but has been noted much more recently—at least within ML community—in its functional extensions, such as in Sec 3.3 of [E].
> - Our decision to focus on the statistical perspective is to provide a perspective might be the most broadly appreciated by the audience with less background in control theory, as is often the case at machine learning conferences based on our previous experience.
> We regard Dynkin's formula as the discrete analogue of Ito's lemma that is widely used in continuous diffusion models. Just as Ito's lemma underpins many works on controllable diffusion models, Dynkin's formula can be used to derive the discrete adjoint and potentially extend to other applications. Nevertheless, we **agree with the reviewer that a control-theoretic derivation has its value**, so we decided to include it in Section 3, which follows the same derivation as adjoint-matching.
>
> &nbsp;
>
> ---
>
> [A] Agapiou, et al. "Importance sampling: Intrinsic dimension and computational cost."
>
> [B] Guo, et al. "Proximal diffusion neural sampler."
>
> [C] Lou, et al. "Discrete diffusion modeling by estimating the ratios of the data distribution."
>
> [D] Liu, et al. "Adjoint Schrödinger bridge sampler."
>
> [E] Park, et al. "Functional adjoint sampler: Scalable sampling on infinite dimensional spaces.".

---

### Official Review · Reviewer_Gkkg · 2025-10-31

**Soundness:** 4
**Presentation:** 4
**Contribution:** 4
**Rating:** 6
**Confidence:** 4

**Summary:**

The paper introduces **Discrete Adjoint Matching (DAM)**, a framework for **fine-tuning discrete generative models**—such as diffusion-based large language models—through an **entropy-regularized optimization** approach. DAM extends the Adjoint Matching (AM) method, previously designed for continuous diffusion models, to discrete state spaces modeled as Continuous-Time Markov Chains (CTMCs).

The authors derive a **discrete analogue of the adjoint system** using Dynkin’s formula, providing a statistical interpretation that avoids the control-theoretic derivations used in AM. This leads to an estimator for the optimal transition rates between discrete states, enabling fine-tuning without requiring differentiability.

To make DAM computationally feasible, the paper introduces **variance reduction** and **importance weighting** techniques, and adapts the method to masked diffusion models, common in language modeling. Theoretically, it provides guarantees via fixed-point and adjoint equations linking DAM to optimal control formulations.

Experiments on synthetic datasets show DAM closely matches the true optimal distributions, outperforming prior baselines like `D1` and `SVDD`. Applied to mathematical reasoning tasks (`GSM8K`, `MATH500`, `Countdown`), DAM yields consistent improvements in accuracy and reward metrics over `D1`.

**Strengths:**

1. **Clear conceptual motivation:**
The paper addresses a timely and well-motivated gap — extending adjoint-based optimization methods, previously limited to continuous diffusion models, to the discrete generative setting, which is crucial for language and symbolic models.
2. **Principled extension of Adjoint Matching:**
DAM is a nontrivial discrete analogue of Adjoint Matching (AM), retaining its optimization-by-simulation philosophy while adapting it to the constraints of discrete-time, discrete-state Markov processes.
3. **Practical algorithmic contributions:**
The inclusion of variance reduction and importance weighting makes the method computationally tractable and numerically stable, which is key for large-scale discrete models.
4. **Bridge between theory and practice:**
By grounding the derivation in both stochastic process theory and discrete optimization, DAM provides a conceptual bridge linking probabilistic control, discrete diffusion, and reinforcement-style objectives.
5. **Empirical effectiveness:**
Experiments show consistent and interpretable improvements over strong baselines (e.g., D1, SVDD) on synthetic reasoning and mathematical datasets, validating the theoretical claims.

**Weaknesses:**

1. **Clarity and depth of the theoretical exposition:**
The theoretical development is solid and well-motivated, but occasionally dense. Some key derivations—particularly the transition from Dynkin’s formulation to the discrete adjoint system—could be presented with more intuition and interpretive discussion, to help the reader understand the underlying mechanics beyond the formal algebra.
2. **Limited discussion of importance sampling techniques:**
The paper briefly introduces importance weighting and variance reduction, which are essential for practical implementation, but this part remains somewhat underexplored.
A more detailed analysis of variance control, proposal distribution selection, or adaptive resampling strategies would strengthen the empirical and theoretical credibility of the approach. It would also help to understand how importance sampling interacts with the entropy-regularized objective in high-dimensional discrete settings.
3. **Positioning within the broader literature:**
The paper’s theoretical contribution could be better contextualized by referencing recent advances connecting stochastic control and diffusion models, such as Pham et al. (2025) [1], where the control–diffusion duality and adjoint computations bear a strong resemblance to those derived here. Acknowledging and comparing these perspectives would clarify the novelty and relevance of DAM within this rapidly developing area.
4. **Empirical diagnostics and ablations:**
Although results are consistent and promising, the paper would benefit from additional ablation studies—for example, isolating the contributions of entropy regularization, importance weighting, and the adjoint update itself. Such diagnostics would offer clearer insight into which components most directly influence performance.

[1] Pham, L.T.N., et al. “_Discrete Markov Probabilistic Models: An Improved Discrete Score-Based Framework with Sharp Convergence Bounds under Minimal Assumptions._” Forty-second International Conference on Machine Learning (ICML, 2025).

**Questions:**

1. **On the discrete adjoint formulation:** Are there particular conditions (e.g., reversibility, bounded rates) under which the discrete adjoint simplifies to known forms?
2. **On importance sampling and variance reduction:**
- The section on importance weighting is promising but rather concise. Could the authors expand on how proposal distributions are selected or adapted during training?
- Can the authors quantify the variance or bias behavior of the estimator as a function of time horizon or dimensionality?
- How sensitive is DAM to poor proposal distributions, and could hybrid approaches (e.g., reweighting combined with entropy regularization) mitigate this issue?
3. **On the link with stochastic control theory:** There are strong conceptual parallels between this work and recent formulations of diffusion models within stochastic control frameworks, such as in Pham et al. (2025) [1]. Could the authors clarify whether DAM can be interpreted as a discrete analogue of the Hamilton–Jacobi–Bellman (HJB) formulation underlying those control-based methods?
4. **On scalability and applicability:**
- How does the computational cost of DAM scale with the number of discrete states or sequence length?
- Could the authors comment on potential approximations or sparsity strategies for handling large vocabularies in text diffusion models?

---

> ### Author Response · Authors · 2025-11-26
> **Response to Gkkg (1/2)**
>
> We thank the reviewer for raising all valuable comments. We are excited that the reviewer identified the significance of our problem, recognized the theoretical soundness of our method, appreciated our algorithmic contribution, and acknowledged effectieness of our empirical results. We believe DAM takes a significant step toward principled methods for fine-tuning discrete generative models. We try our best to resolve all raised concerns in the responses below.
>
> &nbsp;
>
> ---
>
> > **1: Clarification on discrete adjoint theory (W1, Q1)**
>
> - We thank the reviewer for raising these comments. To clarify our presentation in Sec 2 (L153–215), we first note that Lemma 2.1 (Dynkin's formula) provides a principled mathematical tool for estimating function values $f_t(x)$ defined on discrete domains, in a similar vein to Ito's lemma for continuous state spaces. This leads to our main theoretical result, Thm 2.2, which is a straightforward application of Dynkin's formula to $f_t(x; y) := e^{-V_t(y) + V_t(x)}$. For ease of comparison and to highlight both the similarities and differences, we presented Eqs 7, 8, and 3 in similar forms.
> - Despite encouraging, the discrete adjoint in Thm 2.2 is computationally intractable for large discrete state spaces. Instead, we consider an alternative and, crucially, _equivalent_ formulation in Prop 2.3. This analytical solution of the discrete adjoint (Eq 11 in Prop 2.3) does simplify—partially—the discrete adjoint to known forms, as conjectured by the reviewer. Specifically, the first term of the analytical solution of the discrete adjoint (i.e., the red part of $\tilde{a}\_t(y; X_1) = \color{red}{\sum_{z} p^{\text{base}}_{1|t}(z|y)} { \color{red}{e^{-g(z)}} }{ \color{blue}{e^{g(X_1)}} }$) is exactly $e^{-V_t(y)}$. This component is well known in the literature on soft value functions and entropy-regularized RL [A, B].
>
> - Given that the discrete adjoint $\tilde{a}_t(y; X_1)$ is an estimator of $e^{-V_t(y) + V_t(x)}$ and that the aforementioned red part equals $e^{-V_t(y)}$, we can deduce that the remaining blue part $\color{blue}{e^{g(X_1)}}$ is in fact an estimator of $e^{V_t(x)}$. To the best of our knowledge, this has not been presented in the existing literature, and one can prove it formally by applying Dynkin's formula to $f_t(x) := e^{V_t(x)}$ with the dynamics in Eq 50 (*i.e.*, Lemma A.5 in L938). We have included a derivation in Lemma A.6 in L958 of the revision,  and we thank the reviewer for bringing up the topic.
>
> &nbsp;
>
> ---
>
> > **2: Discussion on importance sampling techniques (W2, Q2)**
>
> > **2.1: Proposal distribution**
>
> - Following the clarification in **1**, the discrete adjoint $\tilde{a}_t(y; X_1)$ can be compactly expressed using double expectations: $\tilde{a}_t = \mathbb{E}[e^{-g(z)}]\mathbb{E}[e^{g(X_1)}]$, where the first expectation is taken over the base distribution and the second over the optimal (unknown) distribution.
> - In practice, we apply importance sampling by rebasing both expectations onto the distribution induced by the current model; that is, we use the model distribution as the proposal distribution. This algorithmic design choice is primarily for practical reasons—so that we do not need to simulate twice—and has proven efficient for fine-tuning discrete generative models, as adopted by, e.g., D1 and GRPO [C]. Note that as the model weights change gradually throughout training, so does the proposal distribution.
>
> &nbsp;
>
> > **2.2: Bias and variance**
>
> - As mentioned above Prop 2.4, the discrete-adjoint _without_ importance weighting in Eq 11 exhibits high bias and variance compared to the importance-weighted discrete adjoint in Eq 12. This is mainly due to the opposite signs in the exponential, namely $e^{-g(z)}$ and $e^{g(X_1)}$, which can easily lead to numerical instability.
> - To better illustrate the instability in Eq 11, consider the single-token case with $\mathcal{X} = \\{ \mathfrak{m}, z_{\text{correct}}, z_{\text{wrong}} \\}$, where $g(z_{\text{correct}})=-50$ for $z_{\text{correct}}$ being correct and $0$ otherwise.
> Let the current model $p^u_{1|t}(z_{\text{correct}} | \mathfrak{m})$ be
> $$
> p^u_{1|t}(z | \mathfrak{m}) = \begin{cases}
> \phi, & z = z_{\text{correct}}, \\\\
> 1 - \phi, & z = z_{\text{wrong}}.
> \end{cases}
> $$
> On this example, Eq 11 has a bias of approximately $1-\phi$ and is very large when the model is far from optimal.
> In contrast, Eq 12 suffers from a much smaller bias since it does not follow the original AM recipe and uses importance sampling instead. On the same example, Eq 12 achieves a **much smaller bias** that is approximately $e^{-50}$ smaller than that of Eq 11. We detail the derivation of the biases and variances for this case in Appendix A.5.
> - Nevertheless, we acknowledge that the reduction in variance from Eq 11 to Eq 12 is empirical rather than theoretically driven. This is now clarified in the revised Sec 1 (L117).

---

> > ### Author Response · Authors · 2025-11-26
> > **Response to Gkkg (2/2)**
> >
> > > **2.3: Entropy-regularized objective**
> >
> > - The interaction between importance sampling and the entropy-regularized objective may be best understood through Eq 12. For instance, reducing the contribution of the entropy regularization in Eq 1 is equivalent to increasing the weighting of the terminal cost $g$, which in turn causes the term $e^{-g(Z)}$ to dominate over the KL term $\frac{p^\text{base}(Z|y)}{p^u(Z|y)}$.
> > - The way entropy regularization is reflected in the discrete adjoint can also be explained as follows: *(i)* Given the original entropy-regularized objective, we first rewrite the problem equivalently in terms of $KL(p^u || p^\star)$. *(ii)* Then, in Eq 6, we identify the analytic (albeit intractable) optimal rate $u^\star$ that generates $p^\star$, estimate this rate using the discrete adjoint, and construct the corresponding matching objective.
> >
> > &nbsp;
> >
> > ---
> >
> > > **3: Discussion on Pham et al. (2025) (W3, Q3)**
> >
> > - We thank the reviewer for bringing up this reference. It is exciting to see that a similar mathematical framework—albeit for a different application—has been investigated. We always appreciate the reviewer’s suggestions for positioning our work for a broader audience, and, recognizing it as a highly relevant reference, we are more than happy to include it in the revision.
> > - As conjectured by the reviewer, our DAM is indeed related to the stochastic control perspective using the HJB. In Appendix A.4 of the revised PDF, we have added derivations showing how Eq 85 in Pham et al. (2025) can emerge from the mathematical framework of DAM. While both approaches originate from a similar mathematical foundation, a key difference is that stochastic control frameworks typically aim to solve for the value function $V_t(x)$, whereas our work focuses on the exponential of the value function instead.
> > - From a statistical perspective, this distinction can be viewed as a bias-variance tradeoff: DAM computes an unbiased estimator of the multiplicative correction $e^{-V_t(y) + V_t(x)}$ in the optimal rate, albeit with higher variance (see discussion in **2.2**). In contrast, the value function can typically be estimated with lower variance, but a non-trivial bias is introduced when taking the exponential, due to Jensen's inequality.
> >
> > &nbsp;
> >
> > ---
> >
> > > **4: Clarification on complexity (W4, Q4)**
> >
> > - **Number of discrete states:** We note that the number of discrete states *merely* affects the computational cost of DAM. This is because, for masked diffusion models, the network $u^\theta_t(y, x)$ is parameterized to take $(t, x)$ as inputs and outputs only those $y$ with non-zero probabilities. Practically, the network outputs a logit tensor of size $N \times M$, where $N$ is the sequence length and $M$ is the number of discrete states. Thanks to the high degree of parallelism in modern GPUs, computational complexity scales minimally with $M$.
> > - **Sequence length:** The sequence length does impact the computational cost, as the cost of forward simulation is linear in the sequence length. Fortunately, there are ways to mitigate the cost of forward simulation for longer sequence lengths by using more efficient sampling methods such as Set Block Decoding [D]. As this is orthogonal to the focus of our current work, our method can always be further improved by incorporating such techniques.
> > - **Sparsity in large vocabularies:** As a standard practice, and to ensure consistency with prior works, we did not employ any approximation or sparsity strategies to handle large vocabularies. However, we find that denser reward functions can help make the optimization landscape easier to optimize, especially when large vocabularies—such as those in text—result in a very large space of potential sentences.
> >
> > &nbsp;
> >
> > ---
> >
> > [A] Todorov. "Linearly-solvable Markov decision problems.".
> >
> > [B] Haarnoja, et al. "Reinforcement learning with deep energy-based policies."
> >
> > [C] Shao, et al. "Deepseekmath: Pushing the limits of mathematical reasoning in open language models."
> >
> > [D] Gat, et al. "Set block decoding is a language model inference accelerator."

---

### Official Review · Reviewer_d4QG · 2025-11-01

**Soundness:** 4
**Presentation:** 3
**Contribution:** 4
**Rating:** 8
**Confidence:** 1

**Summary:**

This paper proposes discrete adjoint matching, which is a theoretical framework to conduct reinforcement learning on discrete diffusion models.

**Strengths:**

1, The motivation is clear and significant, locating at the need of reward-guided fine-tuning of discrete diffusion-based models.

2, The theoretical seems to be sound.

**Weaknesses:**

This seems to be a quite good paper. But I am not a theory expert. So I will be alert to any issues raised by other reviewers. Also, I want to raise a question about the performance of Llada-8b on GSM-8K. According to [A], the performance of base Llada model on GSM-8K is 80+. But in your paper, the performance is 60-70. Could you please explain this gap?

Reference:

[A] Revolutionizing Reinforcement Learning Framework for Diffusion Large

**Questions:**

See Weakness.

---

> ### Author Response · Authors · 2025-11-26
>
> We thank the reviewer for raising all valuable comments. We are excited that the reviewer identified the significance of our problem, recognized the theoretical soundness of our method, and appreciated the presentation. We believe DAM takes a significant step toward principled methods for fine-tuning discrete generative models. We try our best to resolve all raised concerns in the responses below.
>
> &nbsp;
>
> ---
>
> > **1: Performance of LLaDA-8B-Instruct on GSM8k**
>
> - We thank the reviewer for raising the comments. We would like to kindly clarify that the paper referenced by the reviewer considers a generation length of 1024, whereas all of our experiments in Sec 5.2—including both our DAM and the D1 baseline—use a generation length of 128, which is an order of magnitude smaller. This difference in generation length, which is known to improve performance [A], largely accounts for the performance gap between the reported values in the two papers.
>
> &nbsp;
>
> - To empirically validate the aforementioned conjecture, we conduct additional experiments on the GSM-8K dataset, increasing the generation length from the originally reported 128 to 256. As shown in the table below, increasing the generation length generally improves performance, with our DAM consistently achieving better test accuracy than the D1 baseline. That said, we believe the results can be further improved with longer generation length, and we thank the reviewer for pointing out the reference.
>
>   | Method | gen_len=128 | 256 |
>   | -------- | -------- | -------- |
>   | LLaDA-8B-Instruct | 68.61 | 76.7 |
>   | $~~$with D1 | 75.59 | 79.8 |
>   | $~~$with DAM (ours) | **75.74** | **79.9** |
>
> &nbsp;
>
> [A] Chen, et al. "Beyond surface reasoning: Unveiling the true long Chain-of-Thought capacity of diffusion large language models."

---

> > ### Comment · Reviewer_d4QG · 2025-11-26
> >
> > Thank you for your response. My concern is addressed. I will keep my 8 score for this good paper.

---

### Official Review · Reviewer_Wztv · 2025-11-01

**Soundness:** 4
**Presentation:** 3
**Contribution:** 3
**Rating:** 8
**Confidence:** 3

**Summary:**

This paper introduces Discrete Adjoint Matching (DAM), which is a novel method for solving the following problem: how to fine-tune discrete generative models (like mask-based diffusion Large Language Models) using the reward-based framework called Adjoint Matching (earlier approach works on continuous diffusion/flow models).

The work is a significant contribution, providing a solution that is not only mathematically elegant and theoretically sound but also demonstrably effective in practice, achieving state-of-the-art results on complex reasoning tasks.

The paper's context is entropy-regularized optimization, which is a formal way to fine-tune a pre-trained generative model ($p^{\text{base}}$) to maximize a reward (or minimize a terminal loss, $g(X_1)$) without straying too far from its original training. The previous method for this, Adjoint Matching (AM), is highly effective for continuous models (like image diffusion models) because it uses gradients ($\nabla g$) as a steering signal. However, this gradient-based approach fails for discrete models because it needs to take a gradient over a discrete set of words. The authors bridge this gap. They propose Discrete Adjoint Matching (DAM), a new formulation that adapts AM to the discrete world. The core innovation is replacing the additive gradient correction of AM with a multiplicative correction factor derived from the exponential of the terminal loss difference (i.e., $e^{-g(y)+g(X_1)}$).

**Strengths:**

- The authors provide a deep theoretical analysis to prove their method for discrete version of AM. They use fixed-point equations to prove that their practical algorithm is guaranteed to converge to the true, theoretically perfect optimal solution
- The authors address a computationally impossible problem in their theoretically optimal solution. They then methodically build a practical solution: estimation via sampling and approximate the correction factor by sampling a few possible futures (K samples). They also adapts the method to masked diffusion models.
- The experiments are clear and decisive. DAM achieves state-of-the-art accuracy on three difficult mathematical reasoning benchmarks

**Weaknesses:**

- The algorithm requires $K$ model-forward passes per training step to build its estimator. While this is clearly effective on an 8B model, the cost for fine-tuning much larger models (e.g., 70B+) is not discussed. A small experiment reporting training time vs. final accuracy for DAM and D1 would make the paper's practical claims much stronger.
- A valuable addition to the empirical analysis would be an ablation study on the number of samples K used in the importance-weighted estimator.

**Questions:**

Please see the weakness above

---

> ### Author Response · Authors · 2025-11-26
>
> We thank the reviewer for raising all valuable comments. We are excited that the reviewer identified the significance of our theoretical contributions, recognized the novelty of our discrete version of AM, and acknowledged our strong empirical results. We believe DAM takes a significant step toward principled methods for fine-tuning discrete generative models. We try our best to resolve all raised concerns in the responses below.
>
> &nbsp;
>
> ---
>
> >**1: Additional experiment and ablation study on number of samples $K$**
>
> - We thank the reviewer for raising these comments. Following the reviewer’s suggestion, we first conduct additional experiment on comparing wall clock times vs final accuracy for DAM and D1. We report the results in the table below and kindly refer the reviewer to Appendix D.1 in the revision for the qualitative plot (Fig 4).
> While the accuracy of D1 plateaus early on, the accuracy of DAM keeps increasing, converging to a much higher accuracy at the end.
>
> |Hours|D1 Accuracy|DAM Accuracy|
> |-----|----------:|-----------:|
> |1|24.35%|17.23%|
> |5|24.70%|24.38%|
> |9|24.98%|23.16%|
> |13|25.05%|32.16%|
> |17|25.89%|69.78%|
> |21|25.80%|85.84%|
>
> &nbsp;
>
> - Next, we conduct an ablation study on how the number of  samples $K$ affects the importance-weighted estimator. As shown in Fig 5 in Appendix D.2, increasing the number of samples improves the convergence speed (due to lower variance) and the accuracy of the converged solution (due to lower bias).

---

### Author Response · Authors · 2025-12-02
**Summary of Discussion Phase**

To help the new AC, we summarize the consensus reached during discussion below:

&nbsp;

## Reviewer Wztv (Original Score: 8)
Reviewer Wztv's raised questions on how the training time vs accuracy compares between DAM and D1, as well as how the choice of the number of samples $K$ affects the convergence speed

In response, we
- Add a new experiment that plots training time vs accuracy between DAM and D1 (Appendix D.1). The accuracy of D1 plateaus early on, the accuracy of DAM keeps increasing, converging to a much higher accuracy at the end.
- Add a new ablation study that studies how the number of samples affects the importance-weighted estimator (Fig 5 in Appendix D.2). Increasing the number of samples improves the convergence speed (due to lower variance) and the accuracy of the converged solution (due to lower bias).

After these additions, the reviewer **raised no further concerns**.

&nbsp;

## Reviewer d4QG (Original Score: 8 -> Maintained)
Reviewer d4QG raised a question on the difference in performance reported by the D1 paper and our work. We clarify that this is due to a difference in generation length, and include a new result that increases the generation length from 128 to 256.

- Reviewer's Quote (Nov 25): "_Thank you for your response. **My concern is addressed**. I will keep my 8 score for this good paper._".

&nbsp;

## Reviewer Gkkg (Original Score: 6)
We appreciate Reviewer Gkkg for raising questions on the relationship of the discrete-adjoint theory to existing work, details on the importance sampling, relationship to Pham et al. (2025) and computational complexity.


In response, we
- Clarify that the numerator of the discrete adjoint relates to known forms in the literature on soft value functions, while the denominator is new
- Illustrate how the use of importances sampling in Eq 12 improves upon Eq 11 by solving the instabilities caused by high bias
- Clarify how DAM can be related to the stochastic control perspective in (Pham et al., 2025) by showing how their framewor
- Clarify that the number of discrete states minimally affects the computational complexity of DAM in terms of the network having a larger output, while the sequence length does affect computational cost but can be allieviated using efficient sampling methods.

After these additions, the reviewer **raised no further concerns**.

&nbsp;

## Reviewer uj3e (Original Score: 6 -> 8)
Reviewer uj3e raised concerns on the convergence of DAM, the correctness of Dynkin's formula and on the control-theoretic presentation. We were able to come to a consensus during the discussion phase, and **Reviewer uj3e raised their score from 6 -> 8**.

- Reviewer's Quote (Nov 26): "_Thank you for clarifying this point. **I agree with you that the original formula is correct**. I thank the authors for their detailed answers... **In light of this improvement, I will slightly increase my score**_"

---

### Meta-Review · Area_Chair_pz1o · 2026-01-06

**Summary:**

The paper proposes Discrete Adjoint Matching (DAM), a method for fine-tuning discrete generative models (specifically Continuous-Time Markov Chains, e.g., masked diffusion models) via entropy-regularised reward optimisation. The authors derive the method using Dynkin’s formula to create a statistical estimator for the optimal transition rate, serving as a discrete counterpart to the "Adjoint Matching" (AM) recently proposed for continuous domains.

Reviewers initially praised the paper for its theoretical novelty, principled derivation, and clear motivation. Their initial concerns focused on:

* The computational cost. Specifically, questions regarding training time vs. accuracy compared to baselines (reviewer Wztv) and scalability to large vocabularies (reviewer Gkkg).
* The discrepancies in reported baseline performance for LLaDA-8B compared to prior literature (reviewer d4QG).
* Some requested theoretical connections. In particular, requests to better contextualise the work within stochastic control literature (e.g., Pham et al., 2025) and clarify the link between the proposed statistical derivation and control-theoretic perspectives (reviewers Gkkg and uj3e).
* The lack of theoretical convergence guarantees (reviewer uj3e).
* And finally, some technical details, such as the stability of the Importance Sampling estimator and the validity of the memoryless assumption (see reviewers Gkkg and uj3e).

The authors provided a comprehensive rebuttal, including new experiments (wall-clock time comparisons, ablation on sample size , increased generation length) and significant text revisions (connection to HJB equations, control-theoretic derivations in the appendix). The rebuttal successfully clarified that baseline discrepancies were due to generation length and demonstrated that DAM converges to higher accuracy than the D1 baseline over time.

I recommend accepting this paper. The authors provided a robust rebuttal that effectively alleviated the identified concerns. New experiments demonstrating Wall-Clock Time vs. Accuracy (Appendix D.1) proved that while DAM is computationally heavier per step, it is more sample-efficient and converges to a higher accuracy than the D1 baseline over time. Theoretically, the authors convincingly argued that masked absorbing states in standard LLMs satisfy the memoryless condition, and they added significant derivations linking their statistical approach to Hamilton-Jacobi-Bellman equations, bridging the gap to control theory. While the empirical gains on standard benchmarks (GSM8K) are modest, the method shows strong performance on combinatorial tasks and, more importantly, offers a principled theoretical advancement for discrete diffusion. Given the strong consensus among reviewers (estimated to be unanimous 8s post-rebuttal) and the high quality of the theoretical exposition, this work merits presentation at ICLR.

**Reviewer Concerns:**

Addressed Concerns:

* *Performance & Efficiency (Wztv):* The authors provided a new comparison of wall-clock time vs. accuracy (Appendix D.1), showing DAM outperforms the D1 baseline over time. An ablation on the number of samples  was also added.
* *Baseline Discrepancies (d4QG):* The authors clarified that the performance gap was due to generation length (128 vs. 1024). New experiments with length 256 showed consistent improvements, fully resolving this concern.
* *Theoretical Context (Gkkg, uj3e):* The authors added discussions linking DAM to the Hamilton-Jacobi-Bellman (HJB) formulation and Pham et al. (2025). They also added a control-theoretic derivation in Section 3, which Reviewer uj3e explicitly appreciated.
* *Importance Sampling (Gkkg):* The authors clarified the bias-variance trade-off between the analytical and importance-weighted estimators.
* *Memoryless Assumption (uj3e):* The authors argued convincingly that pre-trained language models (specifically masked absorbing states) effectively satisfy the memoryless condition required by the theory.

Outstanding Concerns:

* *Convergence Guarantees (uj3e):* While the authors explained that DAM relies on a consistent estimator (SNIS) and shares the lack of strict guarantees with the continuous AM method, a formal convergence proof remains absent.
* *Marginal Gains on Standard Benchmarks:* While the method outperforms D1 significantly on the combinatorial "Countdown" task ( vs ), the gains on GSM8K ( vs ) and MATH500 ( vs ) are relatively marginal, suggesting the method's primary value currently lies in its theoretical contribution and potential for harder search problems rather than immediate SOTA gains on standard math benchmarks.

**Reviewer Scores:**

* **Reviewer Wztv. Original score: 8. Estimated score: 8**. Their request for training time comparisons and ablations was met with new experimental data. The discussion summary notes they raised no further concerns.
* **Reviewer d4QG. Original score: 8. Estimated score: 8**. They explicitly commented during the discussion: "My concern is addressed. I will keep my 8 score for this good paper."
* **Reviewer Gkkg. Original score: 6. Estimated score: 6**. This reviewer gave "Excellent" (4) ratings for Soundness, Contribution, and Presentation in the original review. Their score of 6 appeared to be conditional on clarifications regarding theory and scaling. The authors provided detailed responses and updated the draft with the requested connections to Pham et al. (2025). The discussion summary states the reviewer "raised no further concerns."
* **Reviewer uj3e. Original score: 6. Estimated score: 8**. The reviewer explicitly stated: "In light of this improvement, I will slightly increase my score." Since ICLR 2026 only allows even scores, and the reviewer was "relatively convinced" by the memoryless argument and appreciated the theoretical additions, an 8 is the appropriate estimate.

---

### Decision · Program_Chairs · 2026-01-26

Accept (Poster)